# Influence of Weather Conditions and Climate Oscillations on the Pine Looper *Bupalus piniaria* (L.) Outbreaks in the Forest-Steppe of the West Siberian Plain

**Denis A. Demidko** [1] **, Svetlana M. Sultson** [1] **, Pavel V. Mikhaylov** [1,*] **and Sergey V. Verkhovets** [2]

1  Scientific Laboratory of Forest Health, Reshetnev Siberian State University of Science and Technology, Krasnoyarskii Rabochii Prospekt. 31, 660037 Krasnoyarsk, Russia; sawer-beetle@yandex.ru (D.A.D.); sultson2011@yandex.ru (S.M.S.)
2  The Center for Research and Education "Yenisei Siberia", Siberian Federal University, 660041 Krasnoyarsk, Russia; sverhovec@sfu-kras.ru
*  Correspondence: mihaylovpv@sibsau.ru or mihaylov.p.v@mail.ru

**Abstract:** The pine looper *Bupalus piniaria* (L.) is one of the most common pests feeding on the Scots pine *Pinus sylvestris* L. Pine looper outbreaks show a feature of periodicity and have significant ecological and economic impacts. Climate and weather factors play an important role in pine looper outbreak occurrence. We tried to determine what weather conditions precede *B. piniaria* outbreaks in the southeast of the West Siberian Plain and what climate oscillations cause them. Due to the insufficient duration and incompleteness of documented observations on outbreaks, we used the history of pine looper outbreaks reconstructed using dendrochronological data. Using logistic regression, we found that the factor influencing an outbreak the most is the weather four years before it. A combination of warm spring, dry summer, and cool autumn triggers population growth. Summer weather two years before an outbreak is also critical: humidity higher than the average annual value in summer is favorable for the pine looper. The logistic regression model predicted six out of seven outbreaks that occurred during the period for which weather data are available. We discovered a link between outbreaks and climatic oscillations (mainly for the North Atlantic oscillation, Pacific/North America index, East Atlantic/Western Russia, West Pacific, and Scandinavian patterns). However, outbreak predictions based on the teleconnection patterns turned out to be unreliable. We believe that the complexity of the interaction between large-scale atmospheric processes makes the direct influence of individual oscillations on weather conditions relatively small. Furthermore, climate changes in recent decades modulated atmospheric processes changing the pattern predicting pine looper outbreaks: Autumn became warmer four years before an outbreak, and summer two years before became drier.

**Keywords:** *Bupalus piniaria*; Scots pine; outbreaks prognosis; weather; climate oscillations

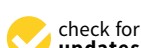

## 1. Introduction

One of the most substantial problems in forestry is large-scale outbreaks of defoliators, for example, repeated outbreaks of *Dendrolimus sibiricus* Tschetverikov in Siberia and the Russian Far East [1,2], *Choristoneura fumiferana* (Clemens) in North America [3,4], and *Zeiraphera griseana* (Hübner) in Western Europe [5]. The defoliation leads to both economic [6,7] and ecological [8] consequences. This challenge is even more meaningful considering that future climate changes can facilitate the onset of outbreaks of phyllophages [9] in both direct and indirect ways.

The pine looper, *Bupalus piniaria* (Linnaeus) (Lepidoptera, Geometridae), is one of the most widespread pests feeding on Scots pine *Pinus sylvestris* Linneus. The species defoliates *P. sylvestris* in late August or early September when the most larvae are in last instars [10]. Pine looper outbreaks occur from 47° N (Austria) to 59° N (Vyatka) and from 2° W (Yorkshire) to 113° E (Transbaikalia) [10,11]. Defoliation of pine stands by *B. piniaria*

often results in high (up to 100%) tree mortality rates [10,12] and causes heavy economic losses [13]. Scots pine is a commercially valuable tree species and plays an important ecosystem role. In this regard, it is essential to study the factors influencing the pest population fluctuations.

The pine looper is a monovoltine species. The flight of the imago in the southern part of Siberia occurs mainly from mid-June to mid-July. Hatching is quite extended: The first larvae appear at the end of June, and the last ones hatch in a month only. The start of hatching coincides with the budburst of Scots pine in the southern part of Siberia [10,14], which is under the control of May temperatures [15]. Until the end of August, the first to third instar *B. piniaria* larvae prevail; in September and October, most are fourth instar or older. In Siberia, the larvae have five instars [10,14], unlike in Europe, where a considerable number of individuals have six instars [16]. By mid-October, almost all individuals migrate to forest litter for pupation and overwintering. The end of October is their deadline to complete the pupation process [10,14]. All the active stages, from reactivated pupae to larvae before they migrate to shelters, under real conditions of the region, are positively dependent on temperature [14,17] and less on humidity [10,17]. Periods with temperatures exceeding the upper limit for this species (about 30 °C) [14] are rare and short [18], and their negative influence on *B. piniaria* populations is negligible. Although data on the effect of humidity on pine looper development are fragmentary, one can assume the higher the air humidity, the more instars are needed before pupation [17].

Outbreaks of forest defoliators depend highly on weather and climate [1,19–23]. By weather, we mean the aggregate of meteorological elements (temperature, precipitation, etc.) at a given moment, while by climate, we mean a long-term weather regime. Outbreaks of pine looper were stated to be linked to regional climate [24] and weather conditions prior to outbreaks [10,11,25]. It is considered proven that drought years contribute to higher abundance of *B. piniaria* [10,11,25]. However, extremely dry conditions, both weather [10] and climatic [24], are also unfavorable for pine looper. In this context, good results should be expected when predicting outbreaks of the pine looper based on the weather data from past years.

Weather conditions depend on oscillations—a large-scale atmospheric circulation that exhibits low-frequency variability over time [26]. Oscillations significantly influence climate in many parts of the globe, including territories distant from the centers of action, where teleconnection indices characterizing oscillations are measured [27]. The relationship between climate oscillations and insect outbreaks has been studied since the beginning of the 21st century ([28]; see review). For forest insects, the effect of oscillations on outbreaks has been shown for *Dendrolimus punctatus* (Walker) [29] and *Z. griseana* [5]. In both cases, the authors stated that oscillations had an impact on insect populations by stimulating weather changes.

Despite numerous arguments for a weather and climate role in provoking defoliator outbreaks, there are a small number of weather-based methods for their prognosis [1,10,30]. Such methods could be helpful to plan control measures early and, correspondingly, reduce defoliation. However, quantitative criteria proposed in these studies are regionally specific, developed based on a small number of outbreaks, and therefore need verification.

The pine looper periodically causes damage to pine forests in Southern Siberia [10,14,31]. It was previously shown that weather is associated with the start of pest outbreaks in South Siberian pine forests [10]. The regional climate is influenced by the processes occurring in other regions of the Earth to some extent [27,32]. We expect that relations in the "oscillations–weather–*B. piniaria*" system are sufficiently strong to use weather, oscillations, or both for pine looper outbreak forecasting. To accomplish the purpose, we considered the following issues (i) what weather conditions precede the pine looper outbreaks and can be used as predictors; (ii) what oscillations affect weather patterns essential to predicting outbreaks; and (iii) whether the oscillations are directly related to pine looper outbreaks. The study was based on the previously performed dendrochronological reconstruction of *B. piniaria* outbreaks [33].

## 2. Materials and Methods

### 2.1. Study Region

The research was carried out in southeast of the West Siberian Plain in the vicinity of Biysk town, Altai Krai (52.52° N, 85.30° E) (Figure 1). The landform is flat [34]. The climate is temperate, somewhat harsh, but warm and humid compared to other parts of the West Siberian Plain [34,35]. Vegetation of the region has undergone significant anthropogenic changes. Prior to agricultural development, the region was dominated by steppe alternating with birch forests [34]. Pine forests in the study region are intrazonal. They cover a small area (several tens of thousands of hectares) along the lower course of the Biya River and the upper reaches of the Ob River.

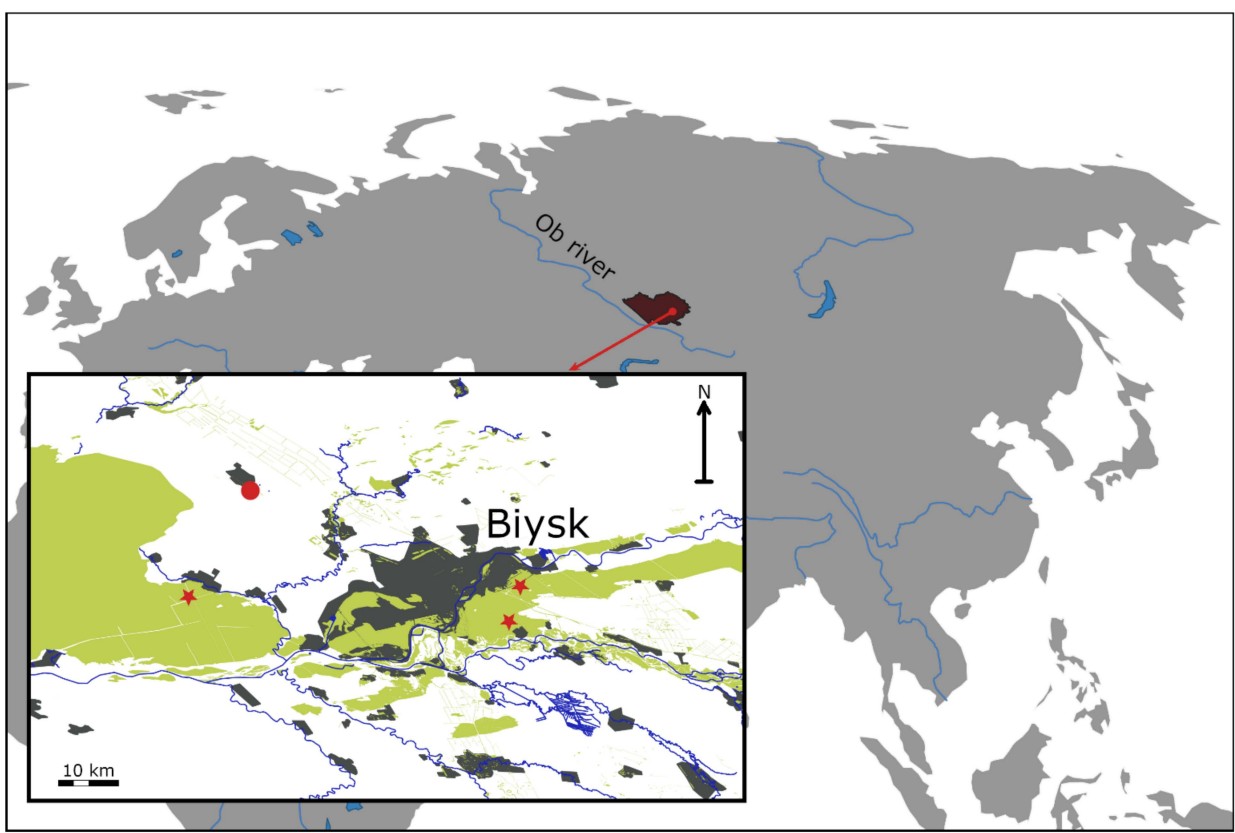

**Figure 1.** Location of the Altai Krai (brown polygon on the Eurasia map), stands (red stars on the inset), and weather station (red circle in the inset). In inset: dark-gray areas—settlements, green areas—forested lands.

### 2.2. Study Objects

We used the results of dendrochronological reconstruction of *B. piniaria* outbreaks to accomplish the present study. We selected three Scots pine-dominated stands for the research (Figure 1). The stands are of *mixtoherbosa* (herb-rich) or *pleuroziosa* (green moss) forest types. None of the stands was subjected to strong anthropogenic impacts (felling, fire, heavy recreational load, etc.). All the stands are close to each other (ca. 5–38 km) and to the only nearby weather station Biysk-Zonal'naya (ca. 18–28 km). The reconstruction methods and results are described in detail in [33]. Briefly, we used standard technics for collecting and processing cores [36], measured ring- and seasonwood width with CDendro [37], and, after crossdating, reconstructed outbreak history using our novel method [33] and three other algorithms [38–40]. Literature data [10,14,31] and data received from the Altai Krai Forest Protection Center confirmed the reconstruction results. The outbreak reconstruction data for the Zarech'e site in [33] were the most exhaustive (see Section 3), so we chose them to compare with the weather and climate data. It was found (within the period

for which instrumental weather data are available) that in the Zarech'e site, seven pine looper outbreaks occurred, which began in 1949, 1959, 1972, 1978, 1986, 1996, and 2001. For the two last outbreaks, the number of female pupae per m$^{-2}$ in Zarech'e or nearby stands was from 4.4 to 7.8 (Altai Krai Forest Protection Center). This corresponds to 75 to 100% defoliation [41]. Unfortunately, the regular monitoring of *B. piniaria* population characteristics (e.g., density of pupae per m$^{-2}$) has not been performed, with exception of the abovementioned outbreak years. Thus, for modeling we used the binary variable with values "year of an outbreak onset" and "not a year of an outbreak onset" as a dependent variable, but not numerical data.

### 2.3. Weather and Climate Data

Weather data were obtained from the Biysk-Zonal'naya meteorological station (52.69° N, 84.68° E) for the period 1936–2017 [18]. The following data were used as predictors: average monthly temperature (*Temp*, °C), monthly precipitation (*Prec*, mm), and the Selyaninov hydrothermal coefficient (*HTC*). Selyaninov hydrothermal coefficient characterizing aridity [42] was calculated using the following formula:

$$HTC = \frac{10 \sum Prec}{\sum Temp} \tag{1}$$

*HTC* was calculated only for months with an average monthly temperature of $\geq 10$ °C. The higher the *HTC* values, the better the humidification conditions; a month with $HTC \leq 1$ is considered arid [42].

To study the relationship between climate oscillations (hereafter, we consider "oscillation" and "teleconnection" synonymous for simplicity) and pine looper moth outbreaks, we used the following indices: Arctic oscillation (AO), North Atlantic oscillation (NAO), Pacific Decadal oscillation (PDO), Pacific/North America index (PNAI), Southern oscillation index (SOI), Scandinavian pattern (SCAND), East Atlantic/Western Russia pattern (EA/WR), Polar/Eurasia pattern (POL), and West Pacific pattern (WP). Monthly average indices were obtained from [43,44].

Preliminary results showed that using the characteristics of individual months as predictors of an outbreak does not give satisfactory results. Therefore, we aggregated weather data and oscillation indices for periods corresponding to stages of *B. piniaria* development in Southern Siberia: reactivation (April–May, *HTC* for May only); flight, oviposition, and hatching (June–July); young larvae feeding (June–August); and old larvae feeding and pupation (September–October, *HTC* for September only) [10,14]. For these periods, we calculated the sums of precipitation and *HTC*, mean values of temperatures and oscillation indices. Since the population growth begins several years before the year *t* when defoliation becomes noticeable [1,10,25,45,46], we considered temperature, precipitation, and *HTC* for several years preceding defoliation ($t-2 \ldots t-5$; a total of 48 predictors).

### 2.4. Statistical Analysis

All analyses were performed with R 4.0.2 software. We used random forest regression (randomForest function, randomForest 4.6–14 package) to select weather characteristics most closely related to outbreaks of *B. piniaria* [47]. The dependent variable was the onset of defoliation (see Section 2.2) in the year *t* in Zarech'e, and the predictors were the characteristics of weather conditions (*Temp*, *Prec*, *HTC*) in previous years $t-2 \ldots t-5$.

The random forest algorithm represents the black box system, meaning it does not allow us to discriminate and describe the criteria for the classification. Therefore, we built an outbreak prediction model using logistic regression (glm function, stats 4.0.2 package). In this model, the probability *P* of an outbreak is estimated using the sigmoid function:

$$P = \frac{1}{1 + e^{-(a_0 + a_1 x_1 + \ldots + a_n x_n)}} \tag{2}$$

where $a_0$—intercept, $a_1 \ldots a_n$—coefficients, $x_1 \ldots x_n$—predictors. At $P > 0.5$, defoliation by *B. piniaria* is expected in the given year. We selected predictors using the step function (stats 4.0.2 package) based on the Akaike information criterion (AIC). We assessed the prediction accuracy using a confusion matrix and the following statistics calculated on its basis (confusionMatrix function, caret 6.0–90 package): accuracy (percentage of correct predictions in total), sensitivity (percentage of correct predictions for outbreaks), and specificity (percentage of correct predictions for the absence of outbreaks) [48]. For multicollinearity estimation and pseudo-$R^2$ calculation, we used performance 0.8.0 package (functions check_collinearity and r2_mcfadden, respectively) [49]. Cross-validation of the developed model was performed using the cv.glm function from the boot 1.3–27 package [50]. ANOVA (anova function, stats 4.0.2 package) was used to determine the statistical significance of the predictors.

We used Ward's method (dist and hclust functions, package stats 4.0.2) to classify weather conditions preceding outbreaks.

We used wavelet analysis to study frequency characteristics of the time series and the coherence between outbreaks (and the associated weather characteristics) and oscillations. We used Morlet wavelets to perform the power spectrum analysis [51]. These algorithms were implemented in the WaveletComp 1.1 package [52] (functions analyze.wavelet for studying frequency characteristics and analyze.coherency for studying coherence). In addition, we used superposed epoch analysis (sea function, dplR package) [53] to study the relationship between outbreaks and oscillations.

## 3. Results

### 3.1. Weather Conditions as Predictors of Outbreaks

The random forest method showed that weather conditions four years before defoliation influenced pest outbreak regimes the most. Pine looper population growth in the $t - 4$ year is somehow influenced by the weather of the whole growing season: spring (April–May) and autumn (September–October) temperature, summer (June–August) precipitation and *HTC*, summer, and autumn (September) aridity. Also important is humid weather during spring (April–May for precipitation, May for *HTC*) in year $t - 3$ and summer (both precipitation and *HTC* of June–August) in year $t - 2$ (Figure 2A).

Using logistic regression, we identified the predictors allowing most accurate prediction of outbreaks. Typically, a pine looper outbreak is preceded by a warm spring, dry summer, and wet and cool autumn in $t - 4$ year, combined with a wet summer in $t - 2$ year (Figure 2B–G). However, the analysis of the obtained results proved the model (for which the predictors were chosen by the step function) (redundant model, R) to be redundant, i.e., it contained an excessive collinear predictor(s) and was overfitted as a result (see Acc, Sens, Spec, and $R^2_{MF}$ value in Table 1). The reason is [54] that the coefficient for $Prec_{JJA}$ ($t - 2$) turned out to be negative (Table 1), while it was positively correlated with pine looper outbreaks (Figure 2G).

The additional analysis showed that redundancy was the data on the humidity in $t - 2$ year: The variance inflation factor of precipitation and *HTC* in this year was extremely high. The model (best model, B) constructed after excluding $Prec_{JJA}$ ($t - 2$) showed a slightly lower predictive ability (see $CV_{adj}$, Acc, Sens, Spec, and pseudo-$R^2$ values in Table 1), but at the same time, it stopped being redundant. An attempt to exclude $HTC_{JJA}$ ($t - 2$) instead of precipitation of the same period decreased forecast accuracy, despite the worse ANOVA results for $HTC_{JJA}$ ($t - 2$) compared to $Prec_{JJA}$ ($t - 2$) (Figure 2F–G); for this reason, the precipitation in $t - 2$ year was excluded from the B model. Notably, $Temp_{SO}$ ($t - 4$) and $HTC_S$ ($t - 4$) demonstrate not very close (Spearman's rho $-0.4$) but statistically significant ($p = 0.0003$) correlation (see Section 4).

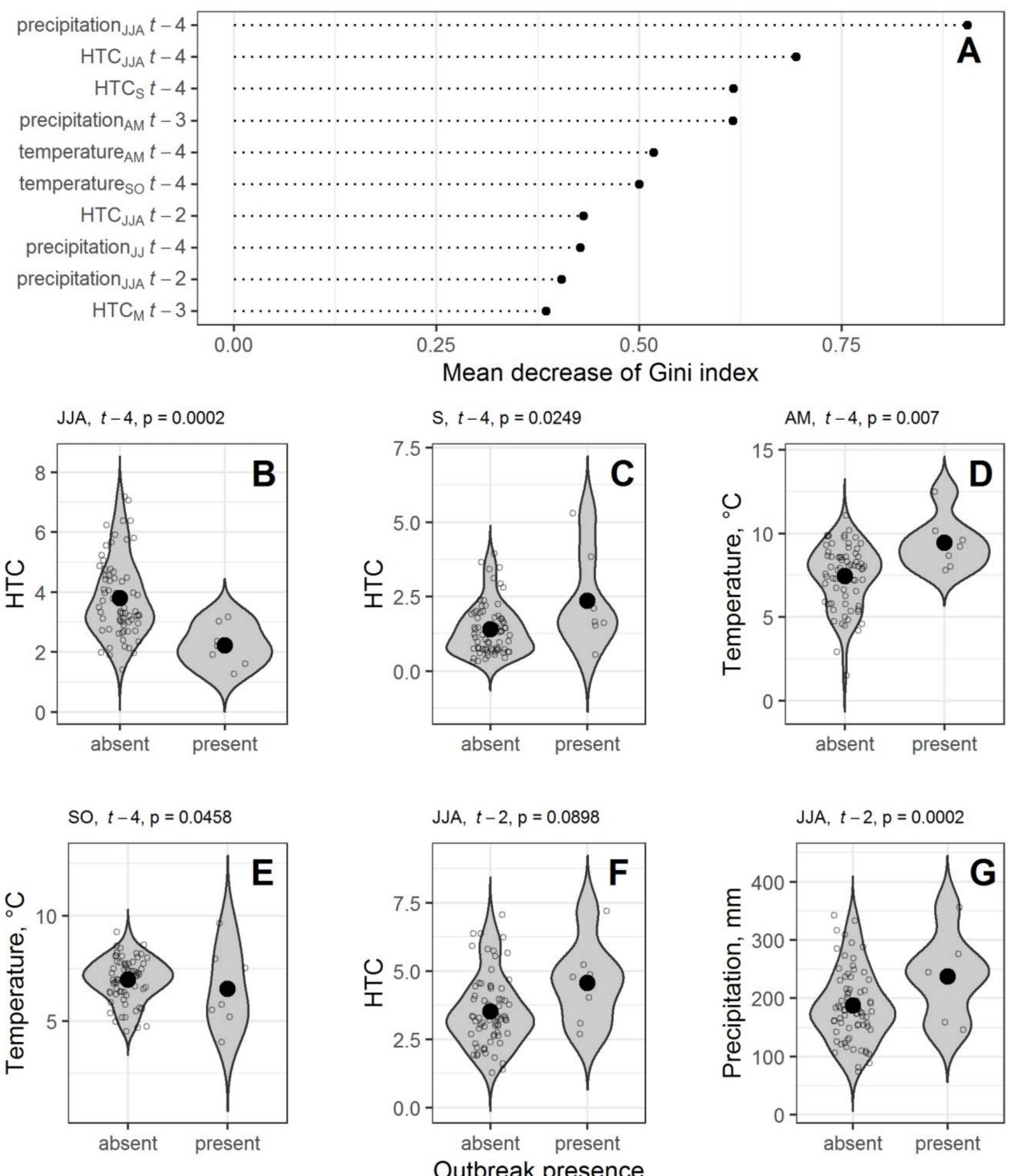

**Figure 2.** Results of selecting predictors for outbreaks of *B. piniaria* using the random forest (**A**) and log-linear analysis (**B–G**). (**B–G**) illustrations for each predictor: months (AM, April and May; JJA, June–August; S, September; SO, September and October), number of years before outbreak ($t − 2$ and $t − 4$, consequently, two and four years before), and significance level of the predictor as determined by ANOVA.

**Table 1.** Logistic regression analysis. For each model, we indicated the predictor coefficients, AIC, adjacent cross-validation estimate of prediction error (CV$_{adj}$), accuracy (Acc), sensitivity (Sens), specificity (Spec), and adjusted McFadden's pseudo-R$^2$ (R$^2_{MF}$).

| Model | Coefficients | | | | | | | AIC | CV$_{adj}$ | Acc | Sens | Spec | R$^2_{MF}$ |
|---|---|---|---|---|---|---|---|---|---|---|---|---|---|
| | Intercept | $HTC_{JJA}$ $t-4$ | $HTC_{S}$ $t-4$ | $Temp_{AM}$ $t-4$ | $Temp_{SO}$ $t-4$ | $HTC_{JJA}$ $t-2$ | $Prec_{JJA}$ $t-2$ | | | | | | |
| R | −744.75 | −63.38 | 63.51 | 143.57 | −81.33 | 380.71 | −7.32 | 14 | 0.058 | 1.000 | 1.000 | 1.000 | 0.957 |
| B | −21.33 | −2.40 | 2.60 | 3.04 | −1.54 | 0.96 | | 26 | 0.091 | 0.974 | 0.857 | 0.986 | 0.659 |

Cluster analysis showed that the weather in the years preceding different outbreaks of the pine looper moth might differ markedly (Figure 3). It seemed most reasonable to distinguish two clusters: 1949, 1959, 1972, 1978, 1996 and 1986, 2001. The second cluster turned out to be too small to use statistical methods when comparing weather characteristics. Therefore, we simply stated the differences between these groups and between them and non-outbreak years (Table 2).

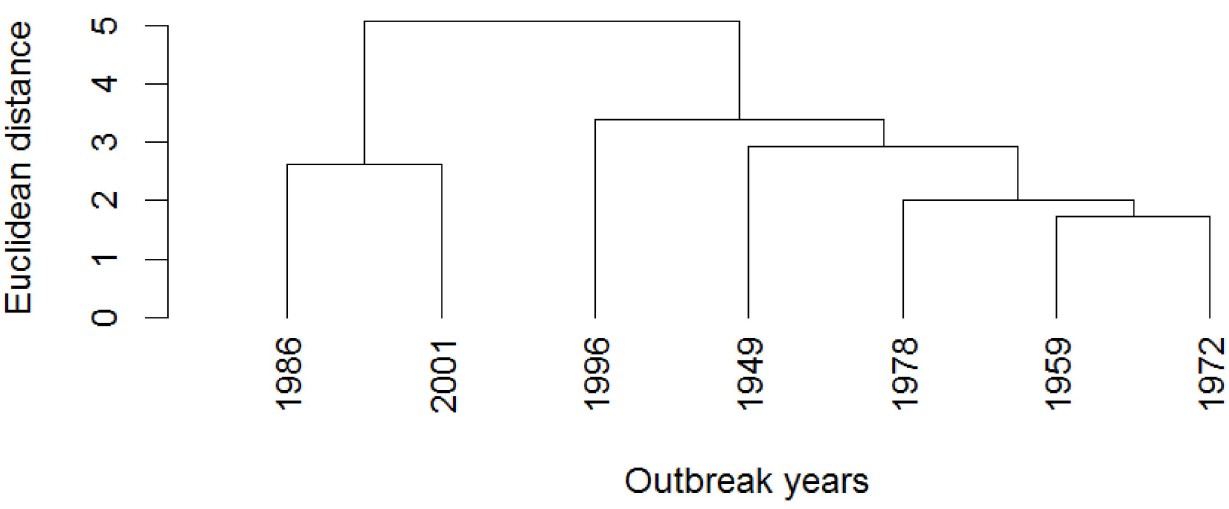

**Figure 3.** Cluster analysis of weather conditions in the pre-outbreak years. The analysis involves the predictors used in model B (Table 1).

**Table 2.** Average values of weather characteristics influencing pine looper outbreaks (*HTC* is dimensionless, °C shows temperature). Weather characteristic symbols as in Table 1 $HTC_{S}$ ($t-4$) are not shown here (see Section 4).

| Years | $HTC_{JJA}$ $t-4$ | $Temp_{AM}$ $t-4$ | $Temp_{SO}$ $t-4$ | $HTC_{JJA}$ $t-2$ |
|---|---|---|---|---|
| 1949, 1959, 1972, 1978, 1996 | 2.09 | 8.66 | 5.70 | 5.22 |
| 1986, 2001 | 2.54 | 11.35 | 8.55 | 2.89 |
| Years with no further outbreaks | 3.80 | 7.44 | 6.95 | 3.52 |

*3.2. Frequency Characteristics of Outbreak Series*

Wavelet analysis of outbreaks for the three studied stands showed their significant similarity (Figure 4). For all series, the most significant maximum average wavelet power falls on a period of ~5 years, which shows good agreement with the data for Bavaria [21]. However, for longer series, significant gaps between time intervals with sufficient wavelet power were noted (mid-1940s to late-1960s for Sokolovo, early 1960s to mid-1970s for Lesnoye). The reason is that these two stands were not damaged in the indicated periods [33]. Therefore, only the data for the Zarech'e site were included in the further analysis.

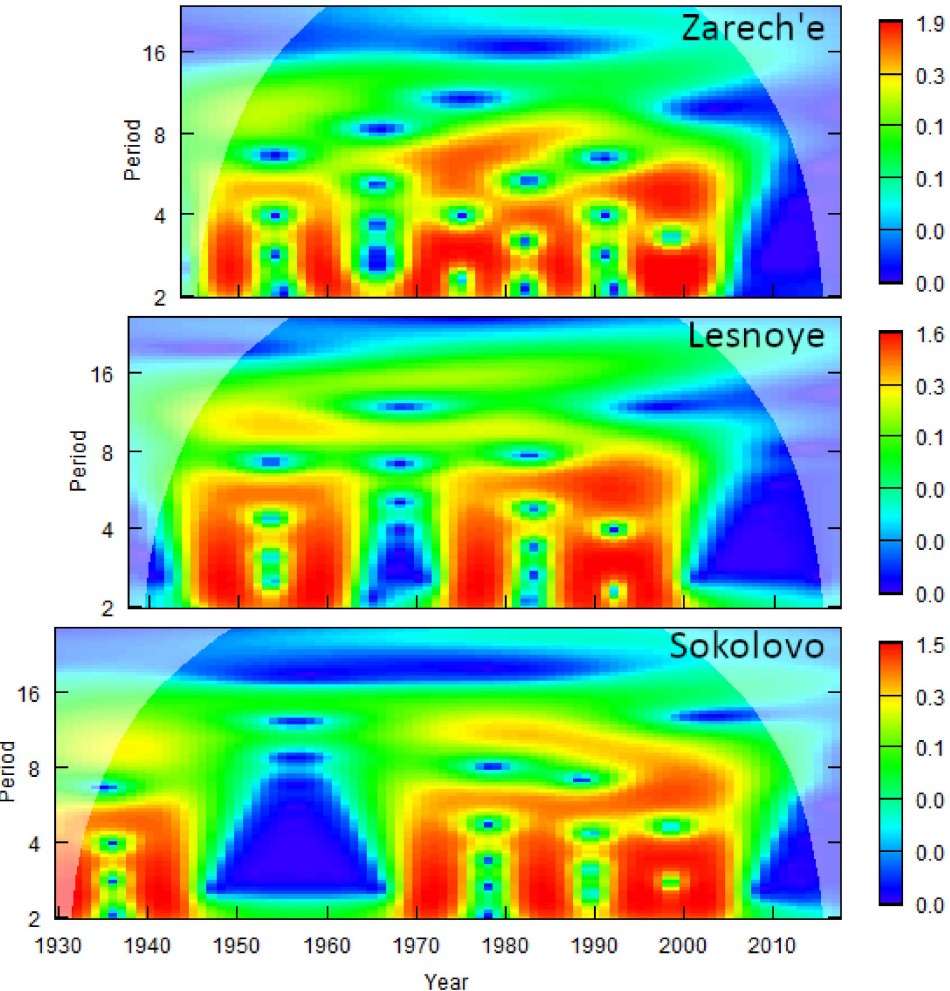

**Figure 4.** Wavelet analysis of the series of pine looper outbreaks reconstructed from dendrochronological data (for the period with the available weather data).

### 3.3. Links between Climate Oscillations, Outbreaks, and Weather

We assessed whether *B. piniaria* outbreaks in the Zarech'e stand and the oscillation indices averaged for different periods of the year were coherent. The assessment showed that each studied oscillation was somehow related to damage to the stands caused by this pest. In most cases, statistically significant coherence between outbreaks and oscillations appeared for periods close to those established by wavelet analysis for outbreak series (around 5 years; see Section 3.2).

Based on the assumption that oscillations affect pine looper outbreaks through weather conditions, we analyzed the coherence of weather characteristics with the oscillation indices for different periods of the year. The analysis confirmed the results obtained in the study of the relationship between oscillations and outbreaks. Namely, a statistically significant coherence ($p = 0.1$) was found for most combinations of "oscillation–weather characteristics–period of the year" (Table 3).

However, superposed epoch analysis showed no statistically significant association between any "oscillation–period of the year" combination and *B. piniaria* outbreaks. Logistic regression with teleconnection indices as predictors and the presence of outbreaks as a response predicted only one outbreak (three with one false-positive prediction if *P* (formula 2) was reduced to 0.4). The predictors selected using a step-by-step procedure based on the Akaike criterion were the SCAND and EA/WR indices for June–August in the $t - 4$ year and the EA/WR indices for the same period in the $t - 2$ year.

**Table 3.** Coherence between climate oscillations and weather conditions–predictors of the pine looper outbreaks. The given periods (4 ≤ period ≤ 10) had the statistical significance of the average wavelet coherence ≤ 0.1. "-" indicates no such periods. Periods intersecting with the range in which the coherence between outbreaks and oscillation was statistically significant at $p = 0.1$ level are shown in bold. If we performed no analysis for "oscillation–weather characteristic" pair (the coherence with the outbreak was weak), the cell remains empty.

| Oscillation | Weather Characteristics | | | |
| --- | --- | --- | --- | --- |
| | *Temp*$_{AM}$ | *HTC*$_{JJA}$ | *Temp*$_{SO}$ | *HTC*$_{S}$ |
| AO | | 4.14, **5.85–6.06**, 6.50 | | |
| NAO | | 4.00, **5.85–6.50** | **6.28–6.50** | 4.00–4.29, 7.46–8.00 |
| PDO | | | - | - |
| PNAI | **4.44–4.76**, 8.88–9.51 | 4.00, **6.06–6.27** | | |
| SOI | 7.73–8.00, 8.88–9.19 | | | |
| SCAND | | 4.00–4.14, **5.66–6.50** | 4.00–4.14, **8.00** | 4.14–4.29, **7.46–8.88** |
| EA/WR | | **8.57–8.88** | **6.06–6.50** | **6.06** |
| POL | 8.88 | | | |
| WP | | **6.06** | 4.00, **5.86–6.06, 6.73–6.96** | **6.06, 7.46** |

## 4. Discussion

### 4.1. Weather Conditions Preceding Pine Looper Outbreaks

Our analysis has shown that spring weather is one of the critical factors influencing insects' development. Spring weather affects both *B. piniaria* and the preimaginal stages of its natural enemies. For example, earlier snowmelt accelerates the adult emergence of *B. piniaria* [14], prolonging caterpillar feeding. However, in our opinion, this factor primarily affects major parasitoids of the pine looper.

The most common species of parasitoids that attack *B. piniaria* in Siberia are *Habrocampulum biguttatum* (Gravenhorst), *Heteropelma megarthrum* (Ratzeburg) (both Hymenoptera: Ichneumonidae: Anomaloninae: Gravenhorstiini), *Blondelia nigripes* (Fallén) (Diptera, Tachinidae, Exoristinae, Eryciini) and *Senometopia pollinosa* (Mesnil) (Diptera, Tachinidae, Exoristinae, Blondeliini) [10]. After wintering, they complete development for a rather long period. The mass emergence of their adults occurs ~10–20 days later than that of the host [14,55,56]. *H. biguttatum* and *H. megarthrum* overwinter in early stages of larval development [55]. It is possible that after wintering, they leave dormancy early to complete their development. Early spring reactivation is also characteristic for *S. pollinosa* [57] and presumably for *B. nigripes* belonging to the same subfamily.

Although air temperature is not the only factor influencing soil moisture during snowmelt [58,59], there is a direct relationship between these two characteristics [59]. Soil moisture increment affects parasitoid larvae that have emerged early, leading to the death of a significant number of them, for instance, two parasitoids attacking sawfly *Pristiphora erichsonii* (Hartig)—tachinid fly *Bessa harveyi* (Townsend) (Diptera, Tachinidae, Exoristinae, Exoristini) and ichneumon *Mesoleius tenthredinis* Morley (Hymenoptera, Ichneumonidae, Ctenopelmatinae) [60]. Tachinid fly *B. harveyi* overwinters in the early stages of larval development, which leads to early reactivation and high sensitivity to a lack of oxygen under high humidity. Ichneumon wasp *M. tenthredinis* overwinters as larvae of the third and fourth instars, which allows them to emerge later and survive in high humidity levels during snowmelt when their metabolic rate is still low. This is consistent with numerous studies, stating that parasitoids are often less resistant to adverse external factors than their hosts [61]. Such data thoroughly explain the relationship between warm spring weather (temperature of April–May for $t − 4$ year) and *B. piniaria* outbreaks (Figure 2): Rapid snowmelt increased soil moisture leading to the death of a significant number of the wintering parasitoids, increasing pine looper survival rate [11].

The second assumed reason for positive influence of April and May temperatures on *B. piniaria* populations is the direct relationship between temperatures and reactivation rate [14]. Nevertheless, we suppose this impact has a minor influence in terms of outbreak

onset because of the prevailing effect of parasitoids on the pupal stage [16]. The third possible reason for such influence is that Scots pine budburst and pine looper larvae hatching are synchronized. A similar effect takes place, for example, in populations of *Z. griseana* (=*diniana*) [62] on European larch and *C. fumiferana* on white spruce and balsam fir [4]. Nonetheless, one should reject this presumption because even young-instar *B. piniaria* larvae consume old needles firstly [10].

Droughts have direct and indirect, often contradictory, effects on insects. The direct effect is associated mainly with temperature regime change. The indirect effect is provided by various changes in food plants (in particular, changes in nutrients and entomotoxins concentrations). Direct and indirect impacts of drought allow plant-eating insects to escape from the control of their natural enemies, causing outbreaks [63]. Although droughts are not always favorable for phyllophagous insects [64], changes are often beneficial to them, at least outside arid regions [65]. Some research reports that dry weather contributes to the outbreaks of *B. piniaria* [10,11].

In Scots pine needles, the concentrations of nutrients increase under drought stress, namely soluble protein fractions [66], soluble carbohydrates, and non-structural carbohydrates in general [67]. Similar studies for other tree and shrub species confirm an increased concentration of soluble carbohydrates and amino acids (in particular, proline) during mild to moderate drought stress [68–70]. However, conflicting data can also be found [71]. Food quality as a factor determining pine looper moth population dynamics was considered earlier [11].

Drought also induces changes in the chemical composition of needles in a negative way for phyllophagous insects. For instance, for *P. sylvestris*, the increase in the content of phenolic compounds in needles was shown under drought conditions [66]. It was discovered for *Diprion pini* (Linnaeus) that an increase in the concentration of phenolic compounds in needles leads to a significant decrease in pupa and adult weight [72], on which the number of eggs per female depends in most insect species [73]. A direct relationship between body weight and fertility has also been described for *B. piniaria* [10], meaning that higher concentrations of phenolic compounds under drought stress negatively affect the number of eggs. However, this factor and the negative impact of low air humidity [10] seem to have a more negligible effect than increased needle nutrient contents. Thus, we associate the contribution of the decrease in summer *HTC* for $t − 4$ year with outbreaks progressing with increased needle nutrient contents. The direct influence of summer temperature on the pine looper larvae development in terms of its outbreaks is negligible (Figure 2).

Autumn weather of $t − 4$ year and summer weather of $t − 2$ year, in contrast to summer of $t − 4$ year, were wet and cool (Figure 2, Table 1), which needs explanation. It has been discovered that extremely low humidity is unfavorable for the pine looper moth larvae [10]. High humidity contributes to a rise in the number of larval instars, leading to increased pupal weight [17]. Since the size of *B. piniaria* pupae is directly related to fertility [10,16], high humidity during larval development in $t − 2$ year contributes to population growth.

It is somewhat more difficult to explain how autumn's high humidity and low temperature in $t − 4$ year contribute to the pine looper population growth. We assume that it is not the increase in *HTC* (which is correlated with temperature) but the decrease in temperature that contributes to population growth. It was indicated for Siberia that a decrease in temperature stimulates larvae to descend from the canopy for pupation and wintering. Larvae that were late to shelter to the forest litter died en masse (up to ~80% of their amount in a crone) from ground frost in autumn [14]. It might also be noted that this factor is not considered to be significant in the milder climate of Western Europe [16,74]. Therefore, it can be assumed that low autumn temperatures stimulate the hibernation of late-instar larvae and thus decrease their mortality from ground frost.

Based on the interpretation of our results, we propose a principal scheme illustrating a *B. piniaria* outbreak development in the south of the West Siberian Plain (Figure 5). It

should be mentioned that under real conditions, the outbreak could begin both earlier and later [33] than shown in the scheme.

**Figure 5.** Principal scheme representing the occurrence of pine looper outbreaks.

However, not every known outbreak has followed this pattern entirely. The most critical factors were warm spring and dry summer in $t - 4$ year. In all cases, the April–May temperature exceeded the average values for non-outbreak years, while *HTC* was lower (Table 2). However, temperatures in September–October in $t - 4$ year before the 1986 and 2001 outbreaks were noticeable higher compared to both other outbreaks and average values for non-outbreak years. The summer in $t - 2$ year for these outbreaks was much drier than would be expected from the above scheme (Figure 5). In addition, the 1986 and 2001 outbreaks were characterized by a warmer spring in $t - 4$ year (Table 2).

Surprisingly, the attempts to use weather conditions in the prognosis of outbreaks of forest phyllophages are pretty rare. Despite the well-known relationship between weather and defoliation by insects (from both Lepidoptera and Hymenoptera) [20,21,25,75], there have been few efforts to develop a weather-based prediction method or at least identify some criteria [1,10,30,76].

The earliest attempts had been established on the simple usage of a single weather characteristic. Due to considerable evidence of drought's dominant role in onset of outbreaks, the *HTC* was chosen as the prognostic criterion. In these prediction systems, droughts in two or more adjacent years are considered triggers of population increase to outbreak level. June and July for *D. sibiricus* [1] and September for *B. piniaria* [10] were proposed as critical periods when insect populations have the highest sensitivity to drought conditions. These prediction methods worked well on training sets but had not been validated. Hence, both methods have a clear biological basis, but their actual predictive ability is unknown. One more reason to use them with care is their simplicity: It looks incredible that only part of the year almost wholly controls population dynamics.

Later a successful attempt to use weather characteristics in spatiotemporal prognosis was made for *C. fumiferana* [76]. The goal of this prediction system differs from ours, making the results rather different too. Most of the predictors in the cited article describe numerical or spatial features of *C. fumeferana* populations. The only weather characteristic (cumulative degree days >5 °C of April) describes the conditions of the critical season (reactivation after wintering), as in the above-cited studies. Nevertheless, it follows the aim of prediction; spatial predictors play leading roles in spatial prognosis.

The approach closest to ours was used to predict *Dendrolimus pini* Linnaeus outbreaks [30]. Both developed prognosis methods are multi-predictor and rely on time intervals that are consistent with stages of insects' life cycles. However, there are some differences between our results. First, *HTC* plays an essential role in *D. pini* activity [77]. Still, mainly temperature and, on a smaller scale, precipitation were included in the decision tree-based prognosis system in [30] as predictors. Second, it is claimed that the onset of *D. pini* outbreaks has been determined by two previous years only. This contradicts

the our assumption that eruptive phyllophagous monovoltine species in forests (like *B. piniaria*) require population growth during three or more years to reach an outbreak level [45]. This discrepancy can be solved by assuming that *D. pini* outbreaks in the Northeastern German lowlands are not eruptive but "prodromal", i.e., simplistic, short-term and not accompanied by decreasing parasitoid control. The development of prodromal outbreaks is more rapid [45].

However, our results are limited in some respects. First, we do not consider the characteristics of stands and their temporal changes. Assuming the correctness of the statement that the most favorable conditions for pine looper outbreaks are in dense, middle-aged (from 40 to 80 years) Scots pine stands [41], we should suppose decreasing of their population with a stand aging. The same refers to the geobotanical features of stands; the most preferable wintering conditions for *B. piniaria* are under the feathermoss layer [10,14]. If the proportion of feathermoss-covered surface decreases (due to fires, logging, or, in the long-term perspective, climate changes), it will lead to the reduction in or even disappearance of pine looper-damaged stands.

Another limitation is related to specific features of *B. piniaria* population from different regions. The traits of weather impact on pine looper population from a distant part of their area can be diverse too. For example, it is suggested that in the southwestern part of the West Siberian Plain and Minusinsk Hollow, the critical precondition of an outbreak is dry August and September [10,78]. In Bavaria, the intensity of damage caused by *B. piniaria* demonstrates a non-linear relationship with winter temperatures [21]. Lastly, our results point out the significant role of low summer humidity and some other weather characteristics (see Section 3). Unfortunately, profound differences in used methods do not allow us to thoroughly analyze the reasons for these discrepancies.

Future climate changes can influence the population dynamics of defoliators positively or negatively. The first alternative is linked to warming and drying of climate [9], and the second one is to discrepancy between animal phenology and changed weather [79]. The example of the last alternative is *Z. griseana* populations in the Alps [5,22]. The cases of the positive impact of a more warm and dry climate are much more numerous, e.g., *Coloradia pandora* Blake in the Cascade Mountains [80], *D. pini* in Northeastern German lowlands [30], and some species in Western and Central Europe [81]. The same idea was proposed in research of temporal dynamics of outbreaks of *Dendrolimus* ssp. complex in the lower reaches of the Yellow River: The wetter the climate, the less area defoliated [20]. For *B. piniaria*, the first scenario is more plausible. The analysis of outbreak activity of this species in Bavaria had shown that outbreaks occurred more frequently during the period of warming (since the 1970s) [21]. The current climate of Siberia shows the tendency to become warmer than earlier [32,82,83]. Hence, it should be reasonable to forecast the increase in pine looper impact on pine forests in Southern Siberia.

*4.2. Influence of Oscillations on the "Weather–Pine Looper" System*

The influence of climate oscillations on Siberia, including the part where our sample plots are located, has been studied mainly in relation to their effect on temperature. In general, we can say for the study area that temperature is related to the ocean temperatures in the regions for which teleconnection indices used here are being calculated [27]. Previously, a link was established between spring, summer, and autumn temperatures of Western Siberia with some teleconnection patterns, in particular, AO, POL, WP, EA/WR, and especially SCAND [32,84]. Except for POL, our results confirm the link between the listed teleconnection patterns, as well as NAO and PNAI, and the weather characteristics used to predict the outbreaks of *B. piniaria* (Table 3). For NAO, this relationship is probably not causal but correlative. For example, the synchronicity of oscillations of AO and NAO is known [85].

Nevertheless, logistic regression and superposed epoch analysis showed no unambiguous relationship between pine looper outbreaks and climate oscillations. We believe that this is due to complex interactions of processes, which are reflected by the considered

teleconnection patterns. For example, there is a link between AO and NAO, SOI with PDO, and NAO [85–87]. A complex system of relationships between different oscillations was described for Europe [88]. When the same weather characteristics depend on several oscillations [32,88], the same result can be achieved with different combinations of their indices. Thus, the direct contribution of individual oscillations to the formation of the weather is relatively small. This makes it difficult to find links between climate oscillations and insect outbreaks. For instance, the study on the role of NAO in the population size dynamics for 49 butterfly species in the United Kingdom showed that this oscillation had a significant effect ($p \leq 0.05$) only on one of the species. For two more butterfly species, the significance of the contribution of NAO to population dynamics was at subthreshold levels ($0.05 \leq p \leq 0.1$) [89].

Attention should also be paid to how weather factors that provoked outbreaks varied over time. Prior to 1986 the situation before the onset of outbreaks matched the scheme proposed above (Table 2, Figure 5), and then later, more or less noticeable deviations began (see the previous subsection). We assume that this is due to global changes in atmospheric circulation. Indeed, it was found for Western Siberia that the strength of the relationship between various circulation mechanisms and weather conditions changes significantly over time [84]. Similar processes are taking place in other regions of the Earth [90,91]. Therefore, we assumed that global changes led to a redistribution of the contribution of individual oscillations to weather and, consequently, to outbreak onset. Such temporal instability additionally complicates the study of the connection between oscillations and the processes that they affect.

Indeed, sometimes there is a change in the relationship between *B. piniaria* outbreaks and the phases of oscillations presumably associated with them. For example, before 1986 (inclusive), outbreaks were preceded by local PNAI maxima in April–May. In contrast, later outbreaks were preceded by local minima (Figure 6A). On the contrary, earlier outbreaks followed the minima of autumn EA/WR values, and after 1986, they came several years after the maxima (Figure 6B). One may assume that changes in the global atmospheric circulation led to a change in the contribution of individual weather characteristics to the formation of conditions favorable for outbreaks. Unfortunately, a detailed analysis of this assumption remains speculative. Neither the data on the pine looper outbreaks in the study region nor the length of the series of teleconnection patterns is insufficient.

Meanwhile, outbreaks of *Z. griseana* are associated with negative NAO values, which cause a decrease in November and winter temperatures [5]. At first glance, this contradicts our conclusions about the absence of a direct link between oscillations and the population dynamics of the species studied. However, the cited research deals with NAO fluctuations over decades. The link between short-term fluctuations in NOA indices and both larch budmoth outbreaks and its population growth that did not reach the outbreak is not evident (see Figure 3 in [5]). We studied the link between *B. piniaria* outbreaks and high-frequency oscillations, which determined the discrepancy between our conclusions and the research cited above.

Other attempts to substantiate the relationship between phyllophagous insects outbreaks and climate oscillations were carried out for El Niño–Southern Oscillation (ENSO) and two insect species: *Choristoneura occidentalis* Freeman in the United States [23] and *D. punctatus* in China [29]. As for *C. occidentalis* research, the authors limited themselves to the assumption of such a link mediated by weather conditions [23]. The China study revealed a direct link between La Niña events (cold phases of ENSO) and *D. punctatus* outbreaks. The last results, however, are not entirely satisfactory. The paper does not describe the methodology for analyzing the link between teleconnection patterns and Masson pine caterpillar outbreaks. Only the results of their comparison are given ([29], Table 3), according to which four out of five outbreaks coincide with La Niña. The authors also ignored one of the recorded outbreaks and did not consider the appearance of La Niña between 1967 and 2009, when neither recorded nor reconstructed outbreaks were known.

Thus, the question of how closely outbreaks of *D. punctatus* are related to ENSO has not been finally resolved.

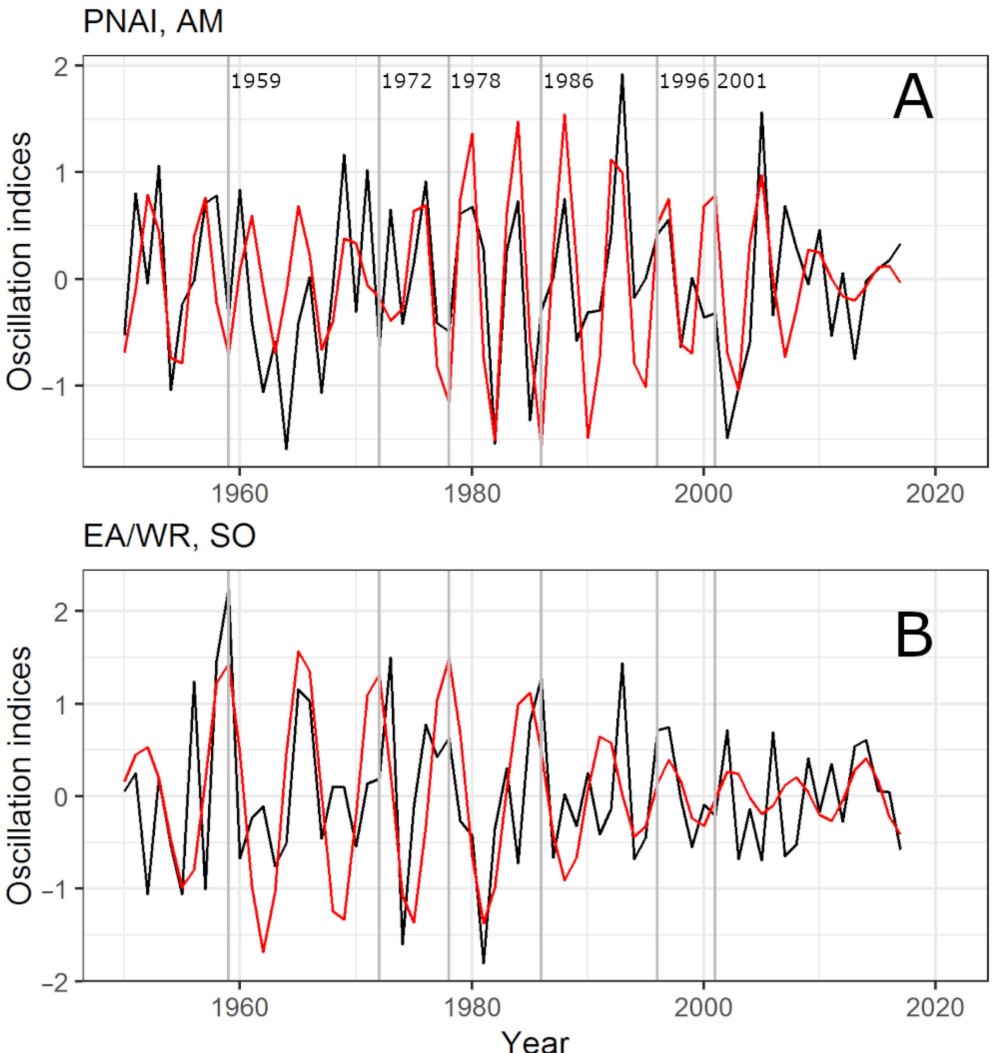

**Figure 6.** Examples of temporal instability of relationship between oscillations ((**A**)—Pacific/North America index, (**B**)—East Atlantic/Western Russia pattern) and outbreaks. Black line—averaged April–May oscillation indices; red line—reconstructed oscillation indices series for period 4 to 10 years (computed by reconstruct function from package WaveletComp 1.1 [52]), and gray vertical lines—starting years of outbreaks except for 1949 (can not be detected by weather).

## 5. Conclusions

Pine looper *B. piniaria* outbreaks in the forest-steppe of Western Siberia occur under the influence of weather conditions of previous years. The most important for their formation are the high temperatures of the middle and late spring and dry summer four years before outbreak onset. Low autumn temperatures four years before an outbreak and a humid summer two years before an outbreak are also significant. The temperature and moisture conditions affect the pine looper directly by reducing its mortality and increasing fertility and indirectly through an increase in the mortality of its parasitoids and a change in the quality of food. Analysis of weather conditions enables us to predict the damage of pine forests caused by this pest accurately. In the last few decades, however, a change in the contribution of individual weather elements to *B. piniaria* population dynamics has been noted.

Assuming that the global atmospheric circulation influences the weather conditions and, consequently, the pine looper population dynamics, we studied the contribution of individual atmospheric oscillations to outbreaks of this species. We determined that, in a number of cases, fluctuations in *HTC* and temperature values for periods of the year that are critical for the pine looper development are coherent with some oscillations (mainly AO, NAO, SCAND, EA/WR, PNAI, WP). However, attempts to directly relate changes in teleconnection patterns to *B. piniaria* outbreaks have generally been unsuccessful. We explain this by the presence of complex nonlinear interdependencies between different atmospheric circulation patterns, which does not allow us to find an unambiguous relationship between them and pine looper population growth. In addition, one cannot exclude the instability of the contribution of individual oscillations to *B. piniaria* outbreaks, associated with changes in the atmospheric circulation since the end of the 20th century.

**Author Contributions:** Conceptualization, D.A.D.; methodology, D.A.D.; formal analysis, D.A.D. and S.M.S.; investigation, D.A.D., S.V.V. and P.V.M.; writing—original draft preparation, D.A.D.; writing—review and editing, D.A.D. and S.M.S.; funding acquisition, P.V.M. All authors have read and agreed to the published version of the manuscript.

**Funding:** The research was carried out within the project "Fundamentals of forest protection from entomo- and fittings pests in Siberia" (No. FEFE-2020-0014) within the framework of the state assignment, set out by the Ministry of Education and Science of the Russian Federation, for the implementation by the Scientific Laboratory of Forest Health.

**Data Availability Statement:** The data presented in this study are available on request from the corresponding author.

**Acknowledgments:** The authors would like to express their gratitude to A.I. Khalaim, Zoological Institute of RAS, Saint-Petersburg, Russia, for his help in selecting literature data on the biology of parasitic Hymenoptera.

**Conflicts of Interest:** The authors declare no conflict of interest. The funders had no role in the design of the study; in the collection, analyses, or interpretation of data; in the writing of the manuscript; or in the decision to publish the results.

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
