# Peer review of "Influence of Weather Conditions and Climate Oscillations on the Pine Looper Bupalus piniaria (L.) Outbreaks in the Forest-Steppe of the West Siberian Plain"

_forests, doi:10.3390/f13010015_

Round 1
Reviewer 1 Report
In this work, the authors studied the relationships between the outbreaks of the pest Bupalus pinaria and climate oscillation using dendrochronological data.
The results obtained are not so convincing and the environmental patterns found do not always correlate with the outbreaks. However, data obtained here are interesting and addresses a gap in the knowledge of this important pest of Pinus sylvestris. The authors' approach is sound and in keeping with current methodologies. Moreover, the manuscript is generally clearly written.
Minor remarks:
Abstract
Line 13: replace “piniaia” with “pinaria”
Introduction
Line 60: replace “Pinus sylvestris L.” with “Pinus sylvestris Linneus”
Lines 84-85: to facilitate the reader, authors should briefly mention the methods to which they refer
Discussion
Line 238-239: “Diptera: Tachinidae: Exoristinae: Blondeliini” please replace with “Diptera, Tachinidae, Exoristinae, Blondeliini” and in the same way across the body of the test. Morover, I suggest to delete subfamily and tribe for all the specie mentioned.
Author Response
Уважаемый коллега!
Благодарим Вас за время и усилия. Все ваши комментарии будут учтены, за небольшим исключением. Мы согласны с тем, что такие линии, как (Diptera, Tachinidae, Exoristinae, Exoristini) и др. Очень длинные. Тем не менее, мы предпочитаем не удалять названия подсемейств и племен, поскольку некоторые из наших предположений основаны на таксономической близости видов паразитоидов. Отсутствие таксономической информации заставит наших читателей искать ее в узкоспециализированных источниках; это может быть сложно для специалистов, не занимающихся систематикой.
С наилучшими пожеланиями,
Денис Демидко и соавторы
Reviewer 2 Report
The authors present a study about the climatic interaction and the outbreak of the Pine Looper. From my understanding, the technical parts of this work were done carefully and correctly. Moreover, I appreciate the effort the authors collecting the data. The research question is a very interesting and original topic. However, this article has some weaknesses that I list below:
- Introduction:
- General overview about insect outbreaks and the importance to study it (specially in a climate change context using international literature is lacking (other lepidoptera species).
- I suggest to add this references:
- Seidl, R., Thom, D., Kautz, M., Martin-Benito, D., Peltoniemi, M., Vacchiano, G., ... & Reyer, C. P. (2017). Forest disturbances under climate change. Nature climate change, 7(6), 395-402.
- Achim, A., Moreau, G., Coops, N. C., Axelson, J. N., Barrette, J., Bédard, S., ... & Montoro Girona, M. (2021). The changing culture of silviculture. Forestry: An International Journal of Forest Research, cpab047.
- Navarro, L., Morin, H., Bergeron, Y., & Girona, M. M. (2018). Changes in spatiotemporal patterns of 20th century spruce budworm outbreaks in eastern Canadian boreal forests. Frontiers in plant science, 9, 1905.
- What are the original elements of this research? Need to study this subject? Put it in values
- Hypothesis are lacking could be improved.
-
- A bit more information about when these trees are defoliated, and which stage of the pine looper is responsible for the defoliations. Overall, a little information about the life stages of pine looper, with some information about the role of climatic factors in each stage, could be interesting.
- Methods:
Study region:
- Presentation of an area with the map could be interesting to visualize (including the climatic variations along the gradient)
Study objects:
- How far were these three stands? What is the views of the authors regarding this distance, if there is any
Weather and climate data:
- Is the same weather data will be used for all the three stands? It would be better to use weather data generated differently for each stands.
- The inclusion of the climatic factors associated with the phenology of Scots Pine is also be preferable. This is because, those factors can be associated with the occurence of outbreak.
Stastistical analysis
- Was this onset same for among all the stands (line 123) ? No clear , though..!`
- Results:
Weather conditions as predictors of outbreaks:
- Was there some population estimation of the pine looper? What was the basis showing the increment in their population?
- Pretty unclear like what does redundant model means for general reader (line 160)
- a bit of more explanation about the selection of the best model
- It will be interesting to see the Akaike weights, and pseudo-R2 as well (table 1).
- It would be interesting to know if there was some significance different between some pairs of years, not for the weather, but for the oscillations of the pine looper.
- It seems to be unclear when we see in the illustration (figure 1). Also, lots of abbreviation in the figure is making reader to get back to the main text to know what that exactly mean? If author had some solution for this, it would be great.
Links between climate oscillations, outbreaks, and weather
- The column heading and the data in the following rows seems to be unaligned with the same column. Authors could improve the format little bit (table 3).
Influence of oscillations on the "weather – pine looper" system
- It can also be interesting if authors had discussed a bit about, how the future climate will mediate the future outbreaks.
- Discussion:
- Less indicated about the limitation of the study. For example, limitation in acquiring the climatic data, like how far were the stations from the stands, and their accuracy for this model.
- There are important elements that must be more strongly highlighted in this paper:
-
International context is lacking. To compare your results with other studies
- Implications for forest management and conservation? Silvicultural practices influence it too?
Lavoie, J., Montoro Girona, M., Grosbois, G., & Morin, H. (2021). Does the type of silvicultural practice influence spruce budworm defoliation of seedlings?. Ecosphere, 12(4), e03506.
- Future research and study limitations. Regeneration has the same vulnerability that mature trees? Lavoie, J., Montoro Girona, M., & Morin, H. (2019). Vulnerability of conifer regeneration to spruce budworm outbreaks in the eastern Canadian boreal forest. Forests, 10(10), 850.
At this stage, I propose the authors to consider these suggestions in a moderate (no major and no minor) revision, and request the editor not to accept the manuscript until and unless the authors make the changes. Congratulations to the authors, they have realized a good job in this original research.
Author Response
Dear colleague!
Thank You very much for Your time, efforts and advices, and especially for references. We hope, our corrections are quite satisfactory.
- General overview about insect outbreaks and the importance to study it (specially in a climate change context using international literature is lacking (other Lepidoptera species).
Brief review with attention to most widespread species from different region is in the lines 36–42.
- What are the original elements of this research? Need to study this subject? Put it in values
We highlighted original feature of our research and value of studied subject in the lines 86–91.
- Hypothesis are lacking could be improved.
Some changes for better clarity have been make in the lines 95–97, 99.
- A bit more information about when these trees are defoliated, and which stage of the pine looper is responsible for the defoliations. Overall, a little information about the life stages of pine looper, with some information about the role of climatic factors in each stage, could be interesting.
Some data about biology of pine looper are in the lines 44–46 and 52–67.
- Presentation of an area with the map could be interesting to visualize (including the climatic variations along the gradient)
- How far were these three stands? What is the views of the authors regarding this distance, if there is any
- Is the same weather data will be used for all the three stands? It would be better to use weather data generated differently for each stands.
For map, You can see Figure 1; there are not any gradients because of little distances between both plots and the only weather station (lines 121 and 122). In addition, this is the reason for use common weather data for all the plots.
- The inclusion of the climatic factors associated with the phenology of Scots Pine is also be preferable. This is because, those factors can be associated with the occurrence of outbreak.
The Scots pine phenology have a little impact on B. piniaria development. However, we have added something about this matter in Discussion section (lines 316–321).
- Was this onset same for among all the stands (line 123) ? No clear , though..!`
Special thanks for this remark; I completely missed it. It checked in lines 368–370.
- Was there some population estimation of the pine looper? What was the basis showing the increment in their population?
- It would be interesting to know if there was some significance different between some pairs of years, not for the weather, but for the oscillations of the pine looper.
Sadly, but it was not with a little exception (lines 132–133). We used the reconstructed data about onset of outbreaks only (lines 134–138).
- Pretty unclear like what does redundant model means for general reader (line 160)
Some more details are in lines 209–211.
- a bit of more explanation about the selection of the best model
You can see about statistical methods in the lines 181–183, and about process of selection in the lines 209–213.
- It will be interesting to see the Akaike weights, and pseudo-R2 as well (table 1).
A little bit about software for McFadden pseudo-R2 You can see in lines 182–183, the pseudo-R2 values are in the Table 1.
We have not shown Akaike weights for the following reasons. First, we got the impression that this characteristic is useful in the case of large number of competing models. Here we have two ones only. Second, calculation of the Akaike weights give us confusing result. The overfitted redundand model get the value close to 1 (about 0.92), and the more reasonable best model, correspondingly, close to 0.
- It seems to be unclear when we see in the illustration (figure 1). Also, lots of abbreviation in the figure is making reader to get back to the main text to know what that exactly mean? If author had some solution for this, it would be great.
We have tried to solve this problem by adding more details to the caption.
- The column heading and the data in the following rows seems to be unaligned with the same column. Authors could improve the format little bit (table 3).
It have been corrected.
- It can also be interesting if authors had discussed a bit about, how the future climate will mediate the future outbreaks.
This discussion is in the lines 432–445.
- Less indicated about the limitation of the study. For example, limitation in acquiring the climatic data, like how far were the stations from the stands, and their accuracy for this model.
The limitations You mentioned here have been discussed above. Some more important limitations considered, e.g., in lines 414–431.
There are important elements that must be more strongly highlighted in this paper:
- International context is lacking.
For improvement of the manuscript it have been used about 25 new papers, mainly international.
- Implications for forest management and conservation? Silvicultural practices influence it too?
- Future research and study limitations. Regeneration has the same vulnerability that mature trees?
Something about role of prediction systems is in lines 88–89.
It should be mentioned we did not regarded the silviculture and regeneration deliberately. Of course, from both the literature and personal communication of practical foresters we know about defoliation of Scots pine cultures by pine looper. On the contrary, pine looper rather unwillingly attack the natural regeneration (literature data). Nevertheless, the data about defoliation young pines by B. piniaria are not numerous, and we did not collected any samples for the purpose of such investigation. Hence, we have not proved opinion in this field; any our speculation about the matter can lead to bigger volume but not to bigger value of the manuscript.
With many thanks and best regards,
Denis Demidko and colleagues
Round 2
Reviewer 2 Report
Authors have considered the majority of my comments and the manuscript has improved significatly in this current version. I have only minor comments before accept this manuscript. I would like that the author add two new references to the introduction to improve the introduction:
Line 38: Lavoie, J. et al. (2021). Does the type of silvicultural practice influence spruce budworm defoliation of seedlings?. Ecosphere, 12(4), e03506.
L42: Hof, A. R., Montoro Girona, M., Fortin, M. J., & Tremblay, J. A. (2021). Using Landscape Simulation Models to Help Balance Conflicting Goals in Changing Forests. Frontiers in Ecology and Evolution, 818.
Congratulations to the author for this nice work,